

# Kinematic and neuromuscular responses to different visual focus conditions in stand-up paddleboarding

João Freitas[1,2,3], Ana Conceição[1,2], Jan Stastny[4], Jorge E. Morais[5,6], Diogo L. Marques[2,3], Hugo Louro[1,2], Daniel A. Marinnho[2,3] and Henrique P. Neiva[2,3]

[1] Department of Sport Sciences, Sport Sciences School of Rio Maior, Santarém, Portugal
[2] Research Center in Sports Sciences, Health Sciences and Human Development (CIDESD), Covilhã, Portugal
[3] Department of Sport Sciences, Universidade da Beira Interior, Covilhã, Portugal
[4] Brno University of Technology, Brno, Czech Republic
[5] Polytechnic Institute of Bragança, Bragança, Portugal
[6] Research Centre for Active Living and Wellbeing (LiveWell), Polytechnic Institute of Bragança, Bragança, Portugal

Corresponding author
Henrique P. Neiva, hpn@ubi.pt

## ABSTRACT

**Purpose:** This study analyzed the kinematics and muscle activity during the stand-up paddleboarding (SUP) under different visual focus points in three conditions: i) eyes on the board nose, ii) looking at the turn buoy, and iii) free choice.

**Methods:** Fourteen male paddleboarders (24.2 ± 7.1 years) performed three trials covering 65 m, and the electromyographic (EMG) activation patterns and kinematic parameters in four cycle strokes for the left and right sides were analyzed. Surface EMG of the upper trapezius, biceps brachii, triceps brachii, tibialis anterior, and gastrocnemius medialis were recorded. The data were processed according to the percentage of maximum voluntary contraction (%MVC). Speed, stroke frequency (SF), stroke length, and stroke index (SI) were analyzed.

**Results:** The speed, SF, and SI ($p < 0.01$, $\eta^2 \geq 0.42$) showed significant variance between conditions, with the free condition achieving the highest speed (1.20 ± 0.21 m/s), SF (0.65 ± 0.13 Hz) and SI (2.25 ± 0.67 m$^2$/s). This condition showed greater neuromuscular activity, particularly in the triceps brachii during both the left (42.25 ± 18.76 %MVC) and right recoveries (32.93 ± 16.06 %MVC). During the pull phase, the free choice presented higher biceps brachii activity (8.51 ± 2.80 %MVC) compared to the eyes on the board nose (6.22 ± 2.41 %MVC; $p < 0.01$), while showing lower activity in the triceps brachii (10.02 ± 4.50 %MVC *vs.* 16.52 ± 8.45 %MVC; $p < 0.01$) and tibialis anterior (12.24 ± 7.70 %MVC *vs.* 17.09 ± 7.73 %MVC; $p < 0.01$) compared to looking at the turn buoy.

**Conclusion:** These results suggest that a free visual focus allows paddleboarders to enhance their kinematics and muscle activation, highlighting the significance of visual focus strategies in improving both competitive and recreational SUP performance.

## INTRODUCTION

Stand-up paddle boarding (SUP) has spiked in popularity over the last decade (*Hibbert, Kaufman & Schmidt, 2023*; *Neiva, Faíl & Marinho, 2020*), and this can be primarily attributed to its relatively easy learning curve (*Schram, Hing & Climstein, 2016b*; *Waydia & Woodacre, 2016*). Unlike traditional surfing, SUP uses a larger board and a paddle, offering superior buoyancy, greater stability, and enhanced stroke efficiency. Upper-body strength, lower-body stabilization, and core muscle activity work in synergy to create forward movement during SUP (*Ruess et al., 2013a*; *Schram, Hing & Climstein, 2016a*). This way, each stroke involves the activation of different muscles, such as the upper trapezius, biceps brachii, triceps brachii, tibialis anterior, and gastrocnemius medialis (*Freitas et al., 2023*; *Ruess et al., 2013a*; *Tsai et al., 2020*). An optimal contribution of these muscle groups is needed to ensure stroke efficiency and balance (*Ruess et al., 2013a*; *Schram, Hing & Climstein, 2016a*).

As scientific interest in SUP continues to grow worldwide, there is a need to understand the biomechanical factors that can potentially influence paddling performance (*Freitas et al., 2023*; *Neiva, Faíl & Marinho, 2020*; *Ruess et al., 2013a*; *Tsai et al., 2020*). Kinematic analysis is important for performance assessment, as it provides insights into stroke mechanics, efficiency, and optimal movement patterns, ultimately influencing speed (*Abellán-Aynés, Alacid & López-Plaza, 2023*; *Vaquero-Cristóbal et al., 2013*; *Abellán-Aynés et al., 2024*). According to earlier research on sprint kayaking and canoeing, it seems to be important to maintain a steady stroke rate for best results because variations might lead to inconsistent propulsion and lower efficiency (*Goreham et al., 2021*). Inter-stroke steadiness is a key determinant of paddling performance in sprint canoeing, with faster race timings being associated with higher steadiness (*Abellán-Aynés et al., 2024*). Furthermore, the change of key kinematic variables such as stroke frequency (SF), stroke length (SL), and stroke index (SI) has been shown to impact performance (*Vaquero-Cristóbal et al., 2013*). Although not commonly reported as other kinematic measures, SI is an important efficiency metric that reflects an athlete's ability to maintain velocity with an optimal stroke length in swimming (*Barbosa et al., 2010*; *Morais et al., 2021*). Research in sprint kayaking has demonstrated that SI follows a similar trend, peaking early in a race and then declining due to fatigue-related reductions in velocity and stroke efficiency (*Vaquero-Cristóbal et al., 2013*). Thus, SI can provide valuable insights into the efficiency of SUP by measuring the effectiveness of each stroke under different conditions (*Abellán-Aynés, Alacid & López-Plaza, 2023*; *Vaquero-Cristóbal et al., 2013*).

While kinematic factors are essential for paddling efficiency, neuromuscular activation patterns also seem to affect stroke performance and technique. Previous research has found different electromyographic (EMG) activation patterns during the paddling stroke cycle (*Freitas et al., 2023*) and when adopting different postures on the board, such as when standing or kneeling (*Tsai et al., 2020*). Furthermore, studies suggested that paddlers with

better performance demonstrate a more efficient SUP stroke than their poorer performance counterparts, possibly due to a more prominent catch angle and longer stroke length and a higher peak power output (*Brown, Lauder & Dyson, 2011*; *Schram, Hing & Climstein, 2016a*; *Schram et al., 2019*). Since kinematic variables influence stroke efficiency, understanding the interplay between attentional focus and movement execution could be important for optimizing SUP technique.

In addition to the biomechanical factors mentioned earlier, attentional focus significantly influences motor performance and learning across various fields, including precision tasks in surgery and sports skills (*Bull et al., 2023*; *Neumann, 2019*). Attentional focus—beyond just visual focus—determines how motor resources are allocated, which in turn could affect paddling efficiency. Previous research has demonstrated that directing attention to different aspects of the movement, such as body segments or external objects, can lead to distinct movement patterns and muscular activation strategies (*Bull et al., 2023*; *Neumann, 2019*). For instance, *Bull et al. (2023)* found that an external focus of attention enhances technique in skilled cricket batters, while *Neumann (2019)* observed that an external attentional focus benefits the movement economy in weightlifters by promoting automatic motor control. However, the influence of attentional focal points on movement patterns and neuromuscular activation during SUP has received limited attention in scientific literature. In the context of SUP, the paddlers may focus on the nose of the board, a turn buoy, or maintain a free gaze across the water, and these choices may influence the paddler's perception, decision-making, and motor execution, shaping their movement patterns and muscle activation strategies.

In the learning process, SUP techniques enhance balance on the board (*Ruess et al., 2013b*), often requiring paddlers to focus on an object or the coastline while paddling. However, there is a gap in research concerning how changes in the attentional focus affect kinematic and muscular participation during SUP performance. Therefore, this study aimed to assess the kinematic and neuromuscular activation in SUP under different attentional focus conditions through visual fixation points. We hypothesize that different attentional orientations could lead to distinct muscle activation levels and kinematic changes in SUP practitioners, ultimately influencing paddling performance across different sides of the stroke cycle.

## MATERIALS AND METHODS

### Participants

Fourteen recreational right-handed male SUP participants (24.0 ± 7.1 years, 1.73 ± 1.22 m, 58.2 ± 15.5 kg, wingspan 1.79 ± 0.87 m, and body mass index 24.2 ± 4.9 kg/m$^2$) volunteered to participate in this study after being instructed on the procedures. Due to the exploratory nature of this study and the effect sizes observed in similar research (*e.g.*, *Freitas et al., 2023*, *Tsai et al., 2020*), a minimum of 12–15 participants should be appropriate to ensure statistical power while keeping data collection and analysis feasible. Nevertheless, *a priori* power analysis was conducted using G*Power 3.1 software, with a moderate effect size ($\eta^2 = 0.25$), an alpha level of 0.05, and a power of 0.80, which indicated that a minimum of 12 participants would be sufficient for detecting meaningful differences. Participants were

only included if they were ≥18 years old and had at least 6 months of SUP experience with regular practice (1–2 times per week). Participants were excluded if they had any medical conditions, injuries, or impairments that could affect paddling performance or compromise their safety. Given this inclusion criterion, the sample consisted of recreational athletes with relatively stable training loads, minimizing the influence of individual training volume and intensity on the study outcomes. Before testing, the participants were informed about the benefits and risks of the investigation and signed an institutionally approved informed consent document. This study was approved by the University of Beira Interior Ethics Committee (CE-UBI-Pj-2022-042), and all the procedures followed the Declaration of Helsinki regarding human research.

## Study design

Before initiating the official trials, each subject performed a 5-min warm-up at a self-selected frequency and intensity while becoming familiar with the equipment (SUP board Itiwit 10′32″5′, paddle Itwit 170–220 cm). The protocol was performed on an inland lake with no current interference, and the participants had to paddle in a straight line (Fig. 1). Before starting data collection, the temperature and wind conditions were analyzed to determine the feasibility of conducting the trials without interference from external factors. All trials were conducted in calm water, free from current interference. All trials were carried out in the wind direction, and daily recordings were obtained, yielding an average wind speed of 3.4 m/s (Gentle Breeze) according to the Beaufort Wind Scale (*National Weather Service, 2022*). Each participant was asked to remain in a bipedal position and start paddling parallelly along a straight line marked with a length of 65 m. This area was demarcated with starting and finishing points and a boundary rope running its entire length.

In this cross-sectional study, each participant performed three stand-up paddling trials, each under a different focus condition, presented in randomized order: i) Condition #1: eyes on the board nose (the participant focused their gaze on the front of the board throughout the trial); ii) Condition #2: looking to the turn buoy (the participant directed their gaze at the buoy, marking the paddling distance's endpoint); iii) Condition #3: free (the participant paddled without a predetermined focal point, allowing for natural gaze behavior) in random order. Each testing session was conducted within a single day per participant, with all trials completed in a 3-h window to minimize the potential of environmental fluctuations. Participants were tested individually to maintain consistent environmental conditions and minimize external distractions. A 30-min passive rest period was maintained between each of the three trials to prevent fatigue from affecting performance.

To standardize paddling intensity, participants were instructed to maintain a comfortable, submaximal speed approximating 70% of their predicted maximum heart rate (*Shookster et al., 2020*). Heart rate was continuously monitored using the Suunto Smartwatch 9 Peak with a Suunto Smart Heart Rate Belt (Suunto, Vantaa, Finland), ensuring that paddling intensity remained between 70% and 75% of maximum heart rate throughout all trials. Data confirmed that there were no significant differences in heart rate
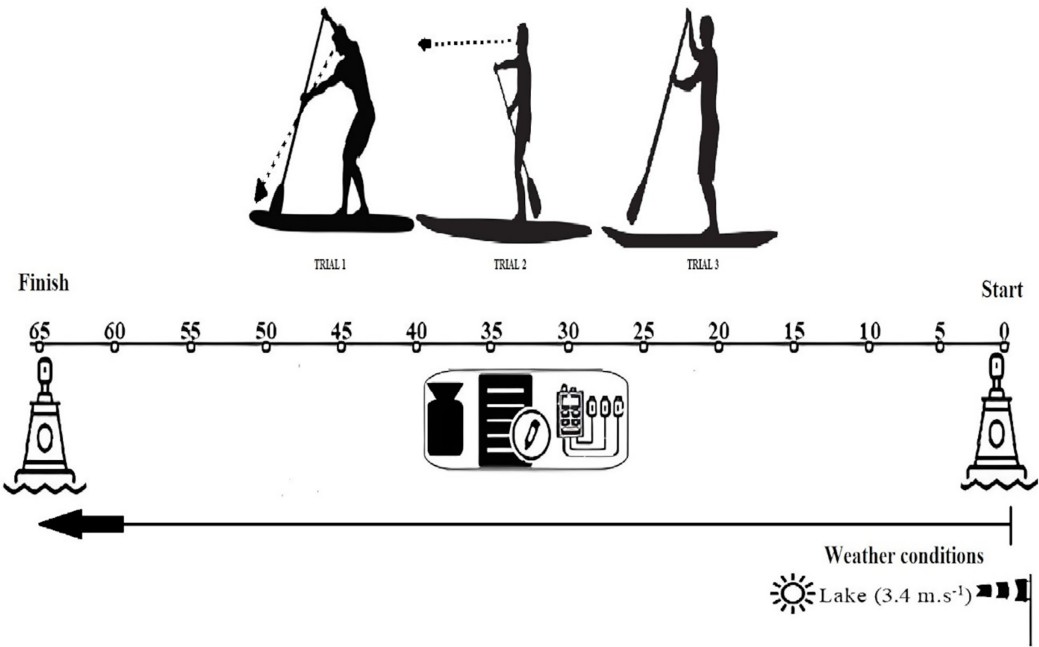

**Figure 1 Scheme of the field data collection protocol of three trials in the different conditions: Trial 1 (Condition #1): eyes on the board nose; Trial 2 (Condition #2): looking to the turn buoy; Trial 3 (Condition #3): free during 65 m at a comfortable speed.**

across conditions (Condition #1: 138.7 ± 16.3 bpm; Condition #2: 141.1 ± 16.9 bpm; Condition #3: 140.6 ± 17.8 bpm; F = 1.26, $p$ = 0.30, $\eta^2$ = 0.09), indicating consistent effort levels between trials.

## Kinematic analysis

The paddling motion was documented through a video system employing a digital video camera (Panasonic, DC-FZ 1000II, 50 Hz, Osaka, Japan) mounted on a tripod (Falcon Eyes, FT-120, Hoogeveen, The Netherlands) perpendicular to the course at 20 m, to capture the entire procedure along the 65-m length. The camera was positioned perpendicular to the paddling trajectory to minimize parallax error and ensure consistent measurement accuracy.

The various stages of the paddling movement were discerned and identified based on predetermined paddling phases (*Freitas et al., 2023*; *Michael, Smith & Rooney, 2009*). The pull phase entails fully immersing the blade in the water and swinging it backward to generate forward power (*Michael, Smith & Rooney, 2009*). The exit and recovery phases involve pulling the blade out of the water and returning it to the starting position before the following catch phase (*Tsai et al., 2020*). A second observer corroborated the event times to ensure precision, enhancing the reliability of measurements associated with the distinct phases executed by the upper limbs.

Following image capture, during the subsequent analysis stage, the Kinovea ® software (version 0.9.5) was used for video editing and conducting kinematic analysis based on sagittal plane images. The selection of the sagittal plane was motivated by its ability to

provide a comprehensive view of the paddling motion, considering its kinematic characteristics. Four cycle strokes were analyzed for each subject's left and right sides. The initial six cycle strokes of each trial and the first stroke of each cycle were excluded to mitigate the impact of the starting acceleration and paddle transfer between sides. The kinematic parameters analyzed in this study included speed, SF, SL, and SI. The speed was derived as distance divided by time, SF (in Hz) as the number of cycles per second, DPS (in m) as the total distance divided by the number of strokes, and SI (in m²/s) as the product of DPS and speed (*Abellán-Aynés, Alacid & López-Plaza, 2023*; *Vaquero-Cristóbal et al., 2013*).

## Surface electromyography

Data were collected as previously described in *Freitas et al. (2023)*. The assessment of muscle activity was conducted on both sides of the body using a wireless EMG system with built-in accelerometers (Miniwave, Cometa, Milan, Italy; EMGandMotionsTools software 8.7.6.0), probes equipped with a 7-gram memory, and a sampling rate of 2,000 Hz at 16 bits. Each subject's skin under the electrodes was shaved, rubbed with sandpaper, and cleaned with alcohol to ensure that the interelectrode resistance did not exceed 5 Kohm (*Afsharipour, Soedirdjo & Merletti, 2019*). Transparent bandages with labels (Hydrofilm®, 10 cm × 12.5 cm, USA) were used to cover the electrodes and isolate them from water (*Hohmann et al., 2006*). The EMG sensors (Kendall $^{TM}$, ECG electrodes, 57 × 34 mm, 57 width mm × 34 length mm, gel area 201 mm$^2$, sensor area 80 mm$^2$, Dublin, Ohio, USA) were placed following the SENIAM recommendations (*Hermens et al., 2000*), and the muscles under analysis were the upper trapezius, biceps brachii, triceps brachii, tibialis anterior, and gastrocnemius medialis, based on their relevance in SUP (*Ruess et al., 2013a*; *Tsai et al., 2020*).

Before the paddle assessment, each subject performed three maximal voluntary isometric contractions on dry land for each muscle analyzed to determine the maximum voluntary contraction (MVC). The MVC test is one of the most used methods of EMG signal normalization (*Castelein et al., 2015*). The contraction was maintained for 5 s, followed by a rest interval of at least 30 s between repetitions. Additionally, a minimum rest period of 1 min was observed before initiating each new test position. The MVC procedures were performed with the examiner applying manual resistance, following the positioning guidelines outlined by both Surface Electromyography for the Non-Invasive Assessment of Muscles (SENIAM) and the Noraxon company (Scottsdale, AZ, USA) (*Al-Qaisi & Aghazadeh, 2015*; *Dyson et al., 1996*). To ensure isometric conditions, the examiner made every effort to adjust the counterforce appropriately. Participants were positioned in a standing posture for the upper trapezius muscle assessment, and the examiner applied a downward force to their shoulders (*Peter, 2005*). Regarding the MVC for the biceps brachii and triceps brachii, these assessments were carried out with the elbow flexed at approximately 90 degrees, as described in previous studies (*Liu et al., 2013*; *Roman-Liu & Bartuzi, 2018*). The examiner stabilized the upper arm to optimize and standardize activation conditions. For the biceps brachii, the forearm was positioned in

supination, while a neutral forearm position was maintained for the triceps brachii. The tibialis anterior was tested with the foot in dorsiflexion, ensuring the toes remained relaxed in a neutral position. For the gastrocnemius medialis, the foot was placed in plantar flexion, prioritizing heel elevation over forefoot pressure. The force was applied against both the forefoot and the calcaneus to achieve maximum pressure in this position, ensuring a pointed, plantar-flexed foot position. The maximum value of the resulting EMG envelope was determined and averaged across the trials for each test (*Boettcher, Ginn & Cathers, 2008*).

During the SUP trials, participants wore custom-made long-sleeved surf suits (Decathlon, Olaian 3/2 mm, Villeneuve-d'Ascq, France) to protect the electrodes and sensors during the trials. During the trials, the EMG measurement was synchronized with a digital video camera (Panasonic, DC-FZ 1000II, Osaka, Japan). Each video was edited according to the trials and then the data was synchronized with the EMG software (EMG and Motion Tools, V8, Cometa, Bareggio Mi, Italy). This study analyzed synchronization by detecting distinct peaks in the accelerometer signal. This approach enabled the identification of the initial and final times of the propulsive phase in the EMG data using the video sequence, achieving an accuracy of 33.3 ms per video frame. The pull and recovery phases of the stroke were determined as previously reported elsewhere (*Freitas et al., 2023*; *Michael, Smith & Rooney, 2009*; *Tsai et al., 2020*). A second observer verified event times to detect any errors. Signal processing commenced with filtering the MVC file. The raw EMG data collected by the sensors underwent initial frequency removal using the following filters: (i) a low-pass filter with a 400 Hz cutoff frequency and a 4th-order Butterworth filter, and (ii) a high-pass filter with a 20 Hz cutoff frequency and a 4th-order Butterworth filter. Each muscle's maximum MVC activation values (μV) were then calculated and presented as percentage MVC (%MVC). The last procedure was to apply the same filters to the signal taken from each trial and the MVC to the trial file. Finally, the mean cycles were exported to an Excel ® file. The muscular activity was presented for the paddling phases of pull and recovery on the left and right sides.

## Statistical analysis

The normality of data distribution was analyzed with the Shapiro-Wilk test. The mean with standard deviation (SD) was calculated as descriptive statistics. An ANOVA for repeated measures, with sphericity checked using Mauchly's test, was used to identify differences between conditions. The total eta square ($\eta^2$) was selected as the effect size index and deemed as: (i) without effect if $0 < \eta^2 < 0.04$; (ii) minimum if $0.04 < \eta^2 < 0.25$; (iii) moderate if $0.25 < \eta^2 < 0.64$ and (iv) strong if $\eta^2 > 0.64$ (*Ferguson, 2009*). The level of significance was set at $\alpha = 0.05$. If necessary, the Bonferroni *post-hoc* correction was used to verify significant differences between pairwise ($p < 0.017$), and Cohen's d estimated the standardized effect sizes: (i) trivial if $0 \leq d < 0.20$; (ii) small if $0.20 \leq d < 0.60$; (iii) moderate if $0.60 \leq d < 1.20$; (iv) large if $1.20 \leq d < 2.00$; (v) very large if $2.00 \leq d < 4.00$; (vi) nearly distinct if $d \geq 4.00$ (*Hopkins et al., 2009*). Statistical analysis was carried out using SPSS statistical software (IBM Corp., Armonk, NY, USA, version 27.0).

## RESULTS

Table 1 presents the descriptive and ANOVA data of the variables measured. Condition #3 (free) presented the fastest speed and SF, and highest SI (Table 1). Specifically, the speed ($p < 0.001$, strong effect), SF ($p < 0.001$, moderate effect), and SI ($p < 0.001$, moderate effect) presented significant variance between groups (Table 1). Regarding the muscular activity, during the right pull, the triceps brachii ($p < 0.001$, moderate effect), biceps brachii ($p < 0.001$, moderate effect), and tibialis anterior ($p = 0.003$, moderate effect) on the right side of the body presented significant variance between conditions. Significant variance was also found during the right recovery for the right upper trapezius ($p < 0.001$, moderate effect), left and right triceps brachii ($p < 0.001$, moderate effects), and right biceps brachii ($p < 0.001$, moderate effect). During the left recovery phase, a significant variance was found in the left and right sides of the upper trapezius ($p < 0.01$, moderate effect) and triceps brachii ($p < 0.01$, moderate effect, and strong effect, respectively).

Regarding kinematics, the Bonferroni correction revealed significant differences between conditions #1 and #3 in speed, SF, and SI. Differences were also noted in speed between conditions #2 and #3 in speed (Table 2). As for muscle activity, the greater differences were found in the left recovery for the right upper trapezius, with greater activity in condition #2 *vs.* condition # 3, and for the right triceps brachii in condition #3 *vs.* condition #1. Moderate effects were found in the remaining differences obtained, with the exception of the activation of the right biceps brachii during the right pull recorded between condition #1 and condition #3. Although the results showed different muscle activity according to each condition, it seems to be clear the higher activation of triceps brachii in the faster condition (condition #3) during recovery phases. During the right pull phase, the biceps brachii showed greater activation in condition #3 (*vs.* condition #1). However, lower activation of triceps brachii and tibialis anterior was found for the faster condition when compared to condition #2.

## DISCUSSION

This study compared the kinematics and neuromuscular activity during SUP between three conditions: i) looking at the board nose, ii) looking at the turn buoy, or iii) free. Significant variations were observed in multiple kinematic and EMG parameters between these conditions. The free choice condition resulted in faster speeds, which may be attributed to the higher SF observed and the highest SI, reflecting enhanced stroke efficiency. Additionally, this condition showed greater neuromuscular activity, particularly in the triceps brachii during both the left and right recovery phases. During the pull phase, this condition revealed higher activation of the biceps brachii (compared with eyes-on-the-board nose condition) and lower activation of the triceps brachii and tibialis anterior (compared to the looking-at-the-buoy condition), indicating a possible shift in muscle recruitment strategies for improved paddling performance.

The higher activation of the triceps brachii during the recovery phase aligns with some previous studies that demonstrated the crucial role of this muscle in paddling motion (*Freitas et al., 2023*; *Tsai et al., 2020*). These studies have shown that the triceps brachii is activated from the late recovery phase through the intermediate pull phase during SUP.

**Table 1 Descriptive statistics of each paddling condition and comparison between the three conditions.** Descriptive statistics of each paddling condition and comparison between the three conditions for speed, stroke frequency, stroke length, stroke index, and muscular activity for the muscles analyzed (presented as percentage MVC), in the pull and recovery phases during the paddling to the right and left side.

| | | Condition #1 Mean ± SD | Condition #2 Mean ± SD | Condition #3 Mean ± SD | F-ratio ($p$) | $\eta^2$ |
|---|---|---|---|---|---|---|
| **Speed (m/s)[a,b]** | | 0.87 ± 0.18 | 0.98 ± 0.23 | 1.20 ± 0.21 | 23.92 (<0.001) | 0.65 |
| **SF (Hz)[a]** | | 0.52 ± 0.09 | 0.54 ± 0.12 | 0.65 ± 0.13 | 9.57 (<0.001) | 0.42 |
| **SL (m)** | | 1.69 ± 0.34 | 1.85 ± 0.32 | 1.86 ± 0.28 | 1.60 (0.224) | 0.11 |
| **SI (m²/s)[a]** | | 1.51 ± 0.56 | 1.83 ± 0.64 | 2.25 ± 0.67 | 9.49 (<0.001) | 0.42 |
| **Right pull (%MVC)** | | | | | | |
| *Upper trapezius* | Left | 20.15 ± 9.27 | 21.79 ± 7.87 | 24.35 ± 8.75 | 4.13 (0.028) | 0.24 |
| | Right | 13.23 ± 7.80 | 14.56 ± 6.17 | 12.28 ± 6.98 | 1.70 (0.202) | 0.12 |
| *Triceps brachii* | Left | 11.99 ± 6.34 | 13.73 ± 7.23 | 10.66 ± 5.10 | 4.88 (0.016) | 0.27 |
| | Right[b] | 12.18 ± 5.58 | 16.52 ± 8.45 | 10.02 ± 4.50 | 10.90 (<0.001) | 0.46 |
| *Biceps brachii* | Left | 9.51 ± 5.27 | 9.39 ± 3.67 | 7.41 ± 3.86 | 5.96 (0.007) | 0.32 |
| | Right[a] | 6.22 ± 2.41 | 6.90 ± 2.66 | 8.51 ± 2.80 | 10.15 (<0.001) | 0.44 |
| *Tibialis anterior* | Left | 12.35 ± 5.96 | 12.88 ± 6.28 | 10.18 ± 6.36 | 2.75 (0.083) | 0.17 |
| | Right[b] | 14.80 ± 7.39 | 17.09 ± 7.73 | 12.24 ± 7.70 | 7.43 (0.003) | 0.36 |
| *Gastrocnemius medialis* | Left | 10.90 ± 5.22 | 10.53 ± 4.33 | 10.98 ± 4.86 | 0.15 (0.858) | 0.01 |
| | Right | 8.04 ± 4.74 | 8.81 ± 4.39 | 8.45 ± 4.06 | 0.43 (0.656) | 0.03 |
| **Right recovery (%MVC)** | | | | | | |
| *Upper trapezius* | Left | 17.28 ± 7.86 | 19.75 ± 9.14 | 14.78 ± 6.37 | 4.73 (0.018) | 0.27 |
| | Right[b] | 14.91 ± 8.19 | 17.38 ± 7.24 | 10.21 ± 5.49 | 15.24 (<0.001) | 0.54 |
| *Triceps brachii* | Left[a,b] | 19.02 ± 10.92 | 21.92 ± 12.50 | 33.57 ± 17.03 | 17.17 (<0.001) | 0.57 |
| | Right[a,b] | 17.66 ± 9.41 | 22.34 ± 12.22 | 32.93 ± 16.06 | 16.00 (<0.001) | 0.55 |
| *Biceps brachii* | Left[a,b] | 7.75 ± 3.40 | 8.17 ± 3.39 | 4.80 ± 2.17 | 20.83 (<0.001) | 0.62 |
| | Right | 8.43 ± 3.86 | 9.16 ± 3.99 | 8.34 ± 5.11 | 0.28 (0.762) | 0.02 |
| *Tibialis anterior* | Left | 18.98 ± 9.25 | 17.84 ± 10.41 | 17.90 ± 11.76 | 0.17 (0.842) | 0.01 |
| | Right | 18.87 ± 11.36 | 19.35 ± 10.11 | 18.70 ± 9.73 | 0.04 (0.963) | 0.01 |
| *Gastrocnemius medialis* | Left | 10.88 ± 4.75 | 10.66 ± 4.26 | 10.43 ± 4.95 | 0.15 (0.865) | 0.01 |
| | Right | 9.43 ± 4.24 | 10.03 ± 5.27 | 10.73 ± 6.85 | 0.50 (0.612) | 0.04 |
| **Left pull (%MVC)** | | | | | | |
| *Upper trapezius* | Left | 11.72 ± 6.37 | 12.72 ± 5.65 | 10.77 ± 5.68 | 1.72 (0.199) | 0.12 |
| | Right | 26.30 ± 13.94 | 26.84 ± 14.52 | 27.60 ± 12.65 | 0.29 (0.751) | 0.02 |
| *Triceps brachii* | Left | 10.21 ± 6.99 | 14.55 ± 9.07 | 11.35 ± 6.63 | 1.74 (0.195) | 0.12 |
| | Right | 11.08 ± 7.63 | 14.30 ± 5.87 | 14.10 ± 6.29 | 2.58 (0.095) | 0.17 |
| *Biceps brachii* | Left | 8.02 ± 3.04 | 8.09 ± 3.43 | 9.24 ± 3.20 | 1.90 (0.170) | 0.13 |
| | Right | 8.35 ± 4.81 | 8.32 ± 4.20 | 7.00 ± 2.68 | 1.68 (0.206) | 0.11 |
| *Tibialis anterior* | Left | 13.50 ± 5.39 | 13.91 ± 7.40 | 13.56 ± 7.78 | 0.03 (0.972) | 0.00 |
| | Right | 15.04 ± 9.30 | 13.21 ± 6.71 | 13.14 ± 7.71 | 0.54 (0.587) | 0.04 |
| *Gastrocnemius medialis* | Left | 9.45 ± 4.00 | 10.04 ± 4.02 | 9.42 ± 3.37 | 0.38 (0.691) | 0.03 |
| | Right | 10.69 ± 5.69 | 11.50 ± 5.27 | 10.43 ± 5.00 | 0.58 (0.568) | 0.04 |
| **Left recovery (%MVC)** | | | | | | |
| *Upper trapezius* | Left[b] | 12.42 ± 5.82 | 14.18 ± 5.20 | 10.21 ± 6.15 | 6.24 (0.006) | 0.32 |

(Continued)

| | | Condition #1 Mean ± SD | Condition #2 Mean ± SD | Condition #3 Mean ± SD | F-ratio (p) | η² |
|---|---|---|---|---|---|---|
| *Triceps brachii* | Right[a,b] | 20.12 ± 9.57 | 20.26 ± 7.68 | 11.85 ± 4.91 | 15.02 (<0.001) | 0.54 |
| | Left[a] | 21.54 ± 11.58 | 25.33 ± 12.49 | 29.96 ± 10.44 | 6.98 (0.004) | 0.35 |
| *Biceps brachii* | Right[a,b] | 19.17 ± 10.89 | 25.56 ± 13.34 | 42.25 ± 18.76 | 31.84 (<0.001) | 0.71 |
| | Left | 10.49 ± 5.14 | 11.63 ± 4.86 | 9.67 ± 7.82 | 0.76 (0.477) | 0.06 |
| *Tibialis anterior* | Right | 5.57 ± 2.80 | 6.59 ± 2.32 | 6.09 ± 4.32 | 0.65 (0.528) | 0.05 |
| | Left | 19.38 ± 9.66 | 18.90 ± 12.23 | 17.67 ± 11.9 | 0.33 (0.719) | 0.03 |
| *Gastrocnemius medialis* | Right | 20.19 ± 13.25 | 21.68 ± 13.33 | 19.59 ± 9.89 | 0.32 (0.731) | 0.02 |
| | Left | 10.59 ± 5.03 | 10.63 ± 3.03 | 12.04 ± 4.91 | 1.62 (0.216) | 0.11 |
| | Right | 9.69 ± 5.42 | 10.68 ± 4.46 | 10.12 ± 5.52 | 0.56 (0.581) | 0.04 |

Note:
Condition #1: eyes on the board nose; Condition #2: looking to the turn buoy; Condition #3: free; SF, stroke frequency; SL, stroke length; SI, stroke index; %MVC, percentage of maximum voluntary contraction; η², eta square (effect size index). Superscript *a*–significant differences between conditions #1 and #3. Superscript *b*–significant differences between conditions #2 and #3.

**Table 2** Pairwise comparison with the Bonferroni *post-hoc* correction and respective effect size (Cohen's d).

| | | Condition #1 *vs.* Condition #3 | | | Condition #2 *vs.* Condition #3 | | |
|---|---|---|---|---|---|---|---|
| | | MD (95 CI) | *p*-value | d (descriptor) | MD (95 CI) | *p*-value | d (descriptor) |
| Speed (m/s) | | −0.33 [−0.43 to 0.23] | <0.001 | 1.69 (large) | −0.22 [−0.37 to 0.07] | 0.005 | 1.10 (moderate) |
| SF (Hz) | | −0.13 [−0.22 to −0.04] | 0.004 | 1.23 (large) | | | |
| SI (m²/s) | | −0.74 [−1.09 to −0.38] | <0.001 | 1.51 (large) | | | |
| **Right pull (%MVC)** | | | | | | | |
| *Triceps brachii* | Right | | | | 6.49 [1.81–11.17] | 0.006 | 0.96 (moderate) |
| *Biceps brachii* | Right | −2.28 [−3.65 to −0.91] | 0.002 | 0.27 (small) | | | |
| *Tibialis anterior* | Right | | | | 4.85 [1.16–8.53] | 0.009 | 0.63 (moderate) |
| **Right recovery (%MVC)** | | | | | | | |
| *Upper trapezius* | Right | | | | 7.16 [4.03–10.30] | <0.001 | 1.11 (moderate) |
| *Triceps brachii* | Left | −14.55 [−23.18 to −5.91] | 0.001 | 1.02 (moderate) | −11.65 [−19.86 to −3.45] | 0.005 | 0.78 (moderate) |
| | Right | −15.27 [−24.63 to −5.92] | 0.002 | 1.16 (moderate) | −10.59 [−17.07 to −4.12] | 0.002 | 0.74 (moderate) |
| *Biceps brachii* | Left | 2.94 [1.30 to 4.59] | <0.001 | 1.03 (moderate) | 3.37 [1.59 to 5.14] | <0.001 | 1.18 (moderate) |
| **Left recovery (%MVC)** | | | | | | | |
| *Upper trapezius* | Left | | | | 3.97 [0.96–6.98] | 0.009 | 0.70 (moderate) |
| | Right | 8.28 [2.62 to 13.94] | 0.004 | 1.09 (moderate) | 8.41 [3.48 to 13.35] | 0.001 | 1.30 (large) |
| *Triceps brachii* | Left | −8.42 [−15.03 to −1.80] | 0.012 | 0.76 (moderate) | | | |
| | Right | −23.08 [−33.56 to −12.61] | <0.001 | 1.56 (large) | −16.69 [−23.16 to −10.22] | <0.001 | 1.06 (moderate) |

Note:
Condition #1: eyes on the board nose; Condition #2: looking to the turn buoy; Condition #3: free; SF, stroke frequency; %MVC, percentage of maximum voluntary contraction; MD, mean difference; 95 CI, 95% confidence intervals.

This activation is primarily driven by the arm extension and trunk rotation required in the paddling motion. The extension of the arms and rotation of the trunk inherently lead to increased muscle activation during SUP paddling (*Freitas et al., 2023*; *Tsai et al., 2020*). Additionally, *Freitas et al. (2023)* found that the triceps brachii on the opposite side of the

stroke typically exhibits higher activation during the pull phase, regardless of paddling direction. This pattern likely reflects the biomechanical demands of stabilizing and driving paddle motion in SUP.

The fixed focus conditions (*e.g.*, looking at the board nose or the turn buoy) were associated with lower kinematic and muscular activity values compared to the free choice condition. This may be due to needing to adjust body positioning and muscular engagement to maintain balance on the board while focusing on a specific point. These adjustments may result in a trade-off, leading to decreased speed and stroke rate but requiring increased stabilization efforts. *Hibbert, Kaufman & Schmidt (2023)* observed that recreational SUP participants tend to rely more on shoulder muscles for propulsion while using their trunk and hip muscles for stability. This dependence on stabilizing muscles might explain the reduced performance metrics observed in fixed-focus conditions, as practitioners prioritize maintaining balance overachieving speed or power output.

It seems that attentional focus, influenced by different visual fixation points, should be essential for maintaining postural control and balance stability. *Schram, Hing & Climstein (2016a)* stated that balance during SUP is significantly affected by vision, with greater postural sway observed when visual input is restricted. These findings correspond with improved performance metrics in conditions that allow for natural and dynamic visual environments. In addition, *Wulf et al. (2010)* and *Marchant, Greig & Scott (2008)* showed that an external focus of attention can enhance neuromuscular coordination and efficiency, leading to better performance outcomes. For instance, *Marchant, Greig & Scott (2008)* discovered that athletes who focused on external targets, such as the bar during a lift, showed reduced biceps brachii activity and improved neuromuscular coordination. Similarly, *Pesce et al. (2007)* demonstrated that athletes with superior visual adjustment skills performed better in tasks requiring dynamic focus shifts, further supporting the benefits of free visual focus. These findings reinforce our results, indicating that allowing a free visual focus can significantly enhance performance metrics, such as speed and muscle activation, in SUP.

The current study suggests that having the freedom to adjust head positioning allows for more natural paddling mechanics, potentially leading to improved stroke efficiency, as indicated by the significantly higher SI observed in the free choice condition. These higher SI values demonstrate the ability to improve the effectiveness of each stroke rather than relying solely on increased SF for faster speeds. Additionally, the results consistently showed higher triceps brachii activation in the faster condition during the recovery phases, emphasizing the importance of this muscle in repositioning the paddle efficiently for the next stroke. During the pull phase, the higher activation of the biceps brachii (compared to the eyes-on-the-board nose condition) and lower activation of the triceps brachii and tibialis anterior (compared to the looking-at-the-buoy condition) further underscore the efficient neuromuscular strategies adopted in the free choice condition. This change in muscle recruitment strategies indicates a more efficient distribution of work among muscle groups. It reduces unnecessary strain on stabilizing muscles, such as the tibialis anterior, while enhancing pulling force. Together, these findings emphasize that improved

kinematics (*i.e.*, SF and SI), optimized muscle activation during recovery phases, and strategic shifts in muscle recruitment during the pull phase are likely key contributors to enhanced paddling performance in the free choice condition.

While this study provides valuable insights into the effects of attentional focus on SUP performance, some limitations should be considered. Cross-sectional design limits the ability to establish causal relationships between attentional focus and performance metrics. Future studies with longitudinal designs could help determine long-term effects. The paddling intensity was self-selected, but heart rate monitoring confirmed consistent effort levels across conditions. Nonetheless, including additional physiological markers could further refine intensity control. Additionally, the sample size and the training level (recreational SUP practitioners) may limit the generalizability of these findings. Expanding the sample to include different experience levels and genders and exploring performance differences under maximal effort conditions would provide a more comprehensive understanding of attentional focus influences on SUP performance.

These findings can potentially influence the development of more effective and targeted training programs and provide valuable insights for improving competitive and recreational performance in SUP. Encouraging paddlers to have a free attentional focus during training can enhance SF, muscle engagement, and overall paddling efficiency. Moreover, the results suggest that SUP training protocols should incorporate different vision fixation strategies to optimize stroke length, paddling speed, and neuromuscular coordination under those conditions. Training under fixed-focus conditions, such as when navigating buoys in races, is crucial for preparing athletes to adapt to varied competitive environments. This dual approach ensures that athletes can balance performance consistency and adaptability across different race conditions. Future studies should examine the causal relationships between attentional focus, neuromuscular activity, and kinematics while also exploring how these findings apply to athletes of various expertise levels and in different environmental contexts.

## CONCLUSIONS

Different attentional focus strategies, influenced by various visual focus conditions during SUP, can significantly affect kinematic performance and neuromuscular activation. The condition allowing paddlers to direct their looks freely resulted in the highest speeds, SI, and SF, indicating improved overall performance. Additionally, this free choice condition showed greater neuromuscular activity, particularly in the triceps brachii during both the left and right recovery phases, emphasizing the critical role of this muscle in paddle repositioning. During the pull phase, the faster condition revealed higher activation of the biceps brachii and lower activation of the triceps brachii and tibialis anterior, perhaps reflecting more efficient muscle recruitment strategies. These findings suggest that a natural and unrestricted attentional focus enables paddlers to adopt more efficient paddling mechanics, likely due to enhanced neuromuscular coordination and reduced unnecessary muscle strain.

### Funding

This work was supported by National Funds by FCT-Foundation for Science and Technology under project number UI/04045 and grant number BIPD/UTAD/7/2023. The funders had no role in study design, data collection and analysis, decision to publish, or preparation of the manuscript.

### Grant Disclosures

The following grant information was disclosed by the authors:
National Funds by FCT-Foundation for Science and Technology: UI/04045 and BIPD/UTAD/7/2023.

### Competing Interests

Henrique P. Neiva is an Academic Editor for PeerJ. The authors declare that they have no competing interests.

### Author Contributions

- João Freitas conceived and designed the experiments, performed the experiments, analyzed the data, prepared figures and/or tables, authored or reviewed drafts of the article, and approved the final draft.
- Ana Conceição conceived and designed the experiments, performed the experiments, analyzed the data, authored or reviewed drafts of the article, and approved the final draft.
- Jan Stastny performed the experiments, analyzed the data, authored or reviewed drafts of the article, and approved the final draft.
- Jorge E. Morais performed the experiments, analyzed the data, prepared figures and/or tables, authored or reviewed drafts of the article, and approved the final draft.
- Diogo L. Marques analyzed the data, prepared figures and/or tables, authored or reviewed drafts of the article, and approved the final draft.
- Hugo Louro performed the experiments, analyzed the data, authored or reviewed drafts of the article, and approved the final draft.
- Daniel A. Marinnho analyzed the data, authored or reviewed drafts of the article, and approved the final draft.
- Henrique P. Neiva conceived and designed the experiments, performed the experiments, analyzed the data, authored or reviewed drafts of the article, and approved the final draft.

### Human Ethics

The following information was supplied relating to ethical approvals (*i.e.*, approving body and any reference numbers):

This study was approved by the University of Beira Interior Ethics Committee (CE-UBI-Pj-2022-042).

## Data Availability

The raw measurements are available in the Supplemental File.

## Supplemental Information

Supplemental information for this article can be found online at http://dx.doi.org/10.7717/peerj.19362#supplemental-information.

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
