# Peer review of "Kinematic and neuromuscular responses to different visual focus conditions in stand-up paddleboarding"

_PeerJ, doi:10.7717/peerj.19362_

## Round 0.1 · original submission · Major Revisions

Both reviewers are positive about the manuscript. However, reviewer 1 in particular has some concerns about the design and methodology that are not easy to address. Please try to resolve these as far as possible or discuss them in the limitations section.

Reviewer 1 ·

Basic reporting

Overall, the idea of the study is goob but there are some major limitations regarding the design of the study.

In addition:

- The manuscript fails to clearly differentiate attentional focus from visual focus, which is crucial for understanding the cognitive processes at play. The hypothesis should focus on attentional focus and its impact on muscle activation and kinematics, rather than misattributing these effects to visual input alone.

- Figures and tables are relevant and appropriately labeled. The raw data is provided, but the methods need further clarification to ensure replicability.

- Most of the references are too old. Please, update them

- The manuscript is self-contained with relevant results, but the major issue with the hypothesis detracts from the overall strength of the conclusions.

- The manuscript generally uses professional and clear English, but some sections could benefit from improved sentence structure for better readability. Minor grammatical adjustments would enhance clarity.

- A more complete and clearer statement of the necessity of your research would be appropriate. Why your study is necessary in the current context of SUP? Include it, please

- In introduction, justify the importance of kinematics and also the importance of stable SRs to performance. In addition, SI is not a very common variable, introduce some references to justify its use. Please, see this references, among others, to do so:

https://pubmed.ncbi.nlm.nih.gov/35575636/

https://pubmed.ncbi.nlm.nih.gov/34013844/

- A paragraph about the study limitations should be included in the discussion section

- The findings of the study should be showed more clearly. What are the implications? How does this study may help to improve things in the future?

Experimental design

-The research question is one of the major limitations as it is not fully clear. It misrepresents the role of visual focus instead of attentional focus.

-The hypothesis should better reflect this distinction to align with existing cognitive theories.

- The second major limitation is regarding the free pace of the paddling. How are you sure the differences in muscle activation and kinematics are due to visual attention and not because they decided to use another intensity along diffferent trials?

- Training intensity, volume and load of the particpants during the week is not indicated and could significantly influence the results. The failure to account for these factors compromises the study's validity.

- Additionally, using a fixed camera angle introduces parallel error, which affects the accuracy of kinematic measurements. Did you have it into consideration? A more precise motion capture system or multiple cameras would have improved data quality for kinematic analysis (and validity).

- What was the speed of the camera? Please, include

- How did you calculate SL, SR and SI? Please, include

- Why did you use 65m? I think, the first section there is an acceleration section

- What statistical program did you use? Please, include

Validity of the findings

- The validity of the results are highly compromised by the design.

- The methodological flaws reduce confidence in the findings.

- The conclusions are based on data that are compromised by lack of control over key confounding variables.

- The study should avoid causal language unless stronger evidence is provided.

- In addition, the small sample size and lack of diversity limit the generalizability of the findings.

Reviewer 2 ·

Basic reporting

The introduction provides relevant information regarding the increasing popularity of Stand-up paddle boarding/SUP, the biomechanical factors influencing performance, and the role of visual focus to different points. The references analyzed in the introduction and in the Discussion section are consistent with the title of the manuscript. The figure and data presented in the two tables provide important information for this investigation.

Experimental design

The research purpose and working hypothesis are well formulated at the end of the introduction. The authors respected the requirements of research involving human subjects. The inclusion and exclusion criteria for the selected participants are clearly stated. The testing conditions, the three different paddling situations compared, the devices and software used in the kinematic analysis and the evaluation of the muscle groups' effort indicate a good organization, a detailed and precise scientific approach to the research topic.

Validity of the findings

The authors' research can be replicated on other groups of subjects, in order to verify the accuracy and completeness of the information provided by this study. The statistical procedures applied are consistent with the experimental design applied in this study and facilitated the obtaining of important results for the multitude of parameters analyzed. The conclusions synthesize the relevant information of the research.

Additional comments

I have attached some suggestions for minor improvements to the initial version of the manuscript:
1. In the Introduction section, it would be useful to include a short paragraph in which to analyze the technical particularities of execution in SUP and the involvement/contribution of different muscle groups in performing the movements specific to this sport.
2. How did you determine the sample size for the group of participants tested?
3. Important information is missing regarding the time interval in which you tested the subjects and the duration of the rest/recovery breaks for the three attempts/variants tested and analyzed.
4. Statistical analysis: It would be useful to indicate the software and version used for the use/application of the parametric techniques described.
5. Table 2: I think you could include the missing data for the empty/blank lines in the table (even if these values did not generate significant differences/you have made a good analysis of statistically significant differences in the text). You could also add the associated data for the pair Condition #1 vs Condition #2.

---

## Round 0.2 · accepted · Accept

All reviewers' comments have been appropriately addressed. I am very pleased to inform you that your manuscript has now been finally accepted and is ready for publication

Reviewer 2 ·

Basic reporting

The authors modified the manuscript according to the suggestions received.

Experimental design

The authors modified the manuscript according to the suggestions received.

Validity of the findings

The authors modified the manuscript according to the suggestions received.

Additional comments

The authors modified the manuscript according to the suggestions received.